# A Global Terrestrial Precipitable Water Vapor Dataset from 2012 to 2020 Based on Microwave Radiation Imager Measurements from Three Fengyun Satellites

Xinran Xia[1], Rubin Jiang[2], Min Min[1], Shengli Wu[3], Peng Zhang[4], Xiangao Xia[5,6]

[1]School of Atmospheric Sciences and Guangdong Province Key Laboratory for Climate Change and Natural Disaster Studies, Sun Yat-sen University and Southern Marine Science and Engineering Guangdong Laboratory (Zhuhai), Zhuhai, China

[2]Key Laboratory of Atmospheric Environment and Extreme Meteorology, Institute of Atmospheric Physics, Chinese Academy of Sciences, Beijing, China

[3]National Satellite Meteorological Centre, Chinese Meteorological Administration, Beijing, China

[4]Meteorological Obsevation Centre, Chinese Meteorological Administration, Beijing, China

[5]LAGEO, Institute of Atmospheric Physics, Chinese Academy of Sciences, Beijing, China

[6]University of Chinese Academy of Sciences, Beijing, China

*Corresponding author*: Dr. Xiangao Xia (xxa@mail.iap.ac.cn)

**Abstract.** A global terrestrial precipitable water vapor (PWV) dataset has been developed using observations from the MicroWave Radiation Imager (MWRI) aboard the FY-3 satellite series (FY-3B, FY-3C and FY-3D) spanning 2012 to 2020. The dataset offers twice-daily PWV records at a spatial resolution of $0.25° \times 0.25°$, aligned with the ascending and descending orbits of the FY-3 satellites. The dataset was generated using an automated machine learning (ML) model that leverages MWRI-based features characterizing surface conditions and an enhanced Global Position System (GPS) PWV dataset as a reference. Trained on over one million sampling points from more than ten thousand stations worldwide, the model ensures a robust representation of global PWV variations. Independent evaluations against SuomiNet GPS and Integrated Global Radiosonde Archive Version 2 (IGRA2) PWV products yielded root mean square error (RMSE) of 4.47 mm and 3.89 mm, respectively, with RMSE values ranging from 2.90 to 5.49 mm across various surface conditions. The dataset effectively captures both spatial and temporal PWV variations, allowing for precise examination of localized and abrupt changes in water vapor induced by extreme weather events. Representing a significant advancement in global terrestrial PWV monitoring, the MWRI PWV dataset provides an all-weather, high-precision data record that bridges gaps in global coverage of passive microwave-based terrestrial PWV observations. It is a

valuable resource for atmospheric research, climate modeling, water cycle studies, and beyond. The

dataset is available at: https://doi.org/10.6084/m9.figshare.26712709 (Xia et al., 2024).

**Key words.** MWRI, PWV, ML, Satellite remote sensing

## 1 Introduction

Atmospheric water vapor plays a prominent role in the Earth system. Not only does it serve as an

important parameter in the global hydrological cycle, but it also dominates the natural greenhouse effect,

accounting for more than 60% of the absorption of atmospheric radiation (Held and Soden, 2000; Wentz

et al., 2007; Xia et al., 2021). Water vapor is essential for accurate weather forecasting and climate change

monitoring (Bojinski et al., 2014; Soden et al., 2002). However, the high spatial and temporal variability

of water vapor poses a significant challenge in maintaining a long-term, precise record of this critical

parameter (Huntington, 2006; Lindstrot et al., 2014; Vogelmann et al., 2015).

Precipitable water vapor (PWV), defined as the total amount of atmospheric water vapor in a vertical

column per unit area, is a primary measure of atmospheric water vapor content (Bedka et al., 2010).

Several techniques have been developed to derive accurate PWV data, with two methods standing out in

particular. The first method is the remote sensing by using Global Positioning System (GPS) signal,

which has been a highly accurate and time-resolved tool for monitoring PWV since the early 1990s

(Bevis et al., 1994; Heise et al., 2009). The second method is radiosonde measurement technique, which

typically provides PWV observations at twice-daily intervals (Durre et al., 2018; Ferreira et al., 2019).

Both methods are highly accurate and are commonly used as benchmarks for validating other PWV

products. In addition to these ground-based measurement methods, satellite remote sensing technique is

increasingly employed for PWV monitoring due to its global coverage (Gao and Kaufman 1992; Du et

al., 2015; Gao et al., 2022). Satellite-based techniques can be categorized into three approaches according

to the spectral bands they used: thermal infrared (TIR), near-infrared (NIR), and microwave (MW)

channels (Gao and Kaufman, 1992; Kaufman and Gao, 2003; Seemann et al., 2003). TIR and NIR

approaches are limited by the presence of clouds and even aerosols, which can obstruct their ability to

retrieve PWV (Du et al., 2015). The NIR approach, while highly accurate, is restricted to daytime

observations due to its dependence on solar radiation reflection. In contrast, the MW approach can

penetrate clouds and aerosols, allowing for PWV observations under all-sky conditions (Gao et al., 2022; Ji et al., 2017; Wang et al., 2009).

PWV retrieval using passive microwave (PMW) imagers is a well-established practice, particularly over oceans. This method has been in operation for decades, resulting in a wealth of operational PWV products now available (Deeter, 2007; Wentz, 1997). However, retrieving PWV using passive microwave (PMW) sensors over land remains challenging due to the weak temperature contrast between the atmosphere and the land surface, as well as the inherent complexity and variability of land surface conditions (Prakash et al., 2018; Wentz, 1997). Several methods have been developed to overcome this difficulty. Deeter (2007) investigated the physical relationship between brightness temperature polarization difference ($\Delta$Tb) signals and PWV (Deeter, 2007). Subsequently, Jones et al. (2010) defined the Microwave Atmospheric Water Vapor Index (MAWVI) based on the ratio of $\Delta$Tb between two microwave frequencies, i.e., 18.7 GHz and 23.8 GHz (Jones et al., 2010). Building upon this, Du et al. (2015) developed a statistical PWV retrieval algorithm for land areas using data from the Advanced Microwave Scanning Radiometer 2 (AMSR2) (Du et al., 2015). The reported root mean square error (RMSE) is 4.7 mm. Similarly, Kazumori et al. (2018) developed a PWV retrieval algorithm that utilized AMSR2 data. The algorithm demonstrated a lower RMSE of 2.4 mm, indicating a further improvement in the accuracy of PWV estimation over land. Note that areas with dense vegetation or ice/snow cover were excluded in these researches due to the specific challenges associated with these surface types. In the presence of dense vegetation, the polarization signals may be insufficiently robust to permit reliable estimation of PWV (Ji et al., 2017; Ji and Shi, 2014). In the case of ice or snow cover, PWV values are typically extremely low, thereby rendering retrieval challenging.

With the development of computer science, and in particular the proliferation of machine learning (ML), has led to the widespread adoption of ML by the remote sensing community (Min et al., 2020). ML is capable of extracting features from vast quantities of data for the purpose of prediction, and is particularly well-suited to addressing nonlinear problems. Gao et al. (2022) proposed an ML method with MAWVI, surface temperature (Ts), cloud liquid water (CLW), and other auxiliary parameters as model features and GPS PWV at approximately 100 stations as the target variable. The accuracy of PWV retrieval over land was further enhanced, with an RMSE of 2.4 mm (Gao et al., 2022). The performance of ML is



contingent upon the diversity and magnitude of the training data, which are indispensable for developing

ML models that generate accurate and unbiased outputs. Using only tens of thousands of training points

from a limited number of stations is unlikely to capture the full range of variability in PWV and its

relationship to other parameters for training a robust model. Thus, the model's performance is likely to

be adversely affected, especially in regions that are underrepresented or not included in the training

dataset. This is confirmed by the fact that stations with an insufficient number of training pairs had

significantly lower accuracy. Xia et al. (2023a) further extend the approach by utilizing 100 million

training data points from tens of thousands of stations across diverse surface conditions to train an ML

model. More importantly, new features were incorporated into the ML model to quantify variations in

surface emissivity, with the goal of extending the retrieval to cover all surface conditions with optimal

accuracy. One of the advantages of this method is that it relies solely on brightness temperature (Tb) data

from the PMW, which paves the way for retrieving PWV from satellite-borne PMW imagers.

The FY-3 series satellites have played a pivotal role in advancing our understanding of the Earth's climate

system and enhancing weather forecasting capabilities since the launch of the inaugural satellite in 2008

(Yang et al., 2012; Zhang et al., 2019). With a total of six satellites launched in three batches, the

microwave payloads and radiometric calibration technology onboard the FY-3 series satellites have

undergone significant evolution over the past decade. Due to temporal instability of radiometers and

variations in instrumentation among different satellites, there is a pressing need to reprocess historical

satellite measurements to ensure the accuracy and stability of satellite data over extended time periods,

which is crucial for establishing Fundamental Climate Data Records (FCDRs) (Wu et al., 2020). In light

of this necessity, the National Satellite Meteorological Center (NSMC) has initiated the recalibration of

Microwave Radiation Imager (MWRI) measurements onboard FY-3 satellites (Li et al., 2022). The

recalibration effort has addressed several early calibration issues, including the emissivity of the hot-load

reflector, intrusion of backlobes from the Earth view, and the nonlinear characteristics of the instruments

(Lawrence et al., 2017; Xie et al., 2021; Yang et al., 2011). Consequently, the precision of MWRI Tb

datasets, particularly in the water vapor absorption channel, has been markedly enhanced. Cross-

comparisons with datasets from other satellites, such as AMSR2 and GMI, validating the effectiveness

of the recalibrated MWRI Tb datasets (Xia et al., 2023b). The comprehensive exploitation of MWRI Tb

data offers considerable potential for advancing climate change research, enhancing numerical weather

forecasting, and fortifying environmental monitoring endeavors. It is anticipated that the utilisation of

these data sources across critical domains will yield significant insights and improvements (Wei et al.,

2022).

In this work, we have created a brand new operational global terrestrial PWV product based entirely on

FY-3 MWRI Tb data for the first time. This dataset fills a gap in PMW-based water vapor products over

only limited land types. Additionally, it addresses a separate but related issue in IR-based water vapor

products that are only available under clear sky conditions. This will facilitate the acquisition of more

accurate water vapor observations under all sky conditions and over all land types, which is expected to

enhance our comprehension of the global atmospheric water cycling.

## 2 Data and methods

### 2.1 FY-3 MWRI Level 1C Tb

The MWRI onboard FY-3 series receives radiance from the Earth's surface and atmosphere at a zenith

angle of 53.1°. The MWRI operates at five frequencies (10.65 GHz, 18.7 GHz, 23.8 GHz, 36.5 GHz, and

89 GHz) with both horizontal and vertical polarization (10 channels), providing valuable opportunities

for remote sensing of atmospheric and surface parameters (Yang et al., 2012; Zhang et al., 2019).

Following the extensive reprocessing of FY-3 historical data, a new version of the long-term recalibrated

FY-3 MWRI L1C Tb dataset has been released by NSMC (Wu et al., 2023). The results of the evaluation

demonstrate that the new recalibrated dataset exhibits a notable enhancement in accuracy, with the RMSE

of each channel remaining below 2 K (Xia et al., 2023b). The NSMC has made available nearly a decade's

worth of FY-3 MWRI L1C Tb data, encompassing the FY-3B (2012-2013), FY-3C (2014-2017), and

FY-3D (2018-2020) periods.

### 2.2 GPS remote sensing and IGRA2 radiosonde PWV data


The enhanced GPS PWV dataset (enGPS PWV) utilized in this study comprises instantaneous PWV

values from 12,552 GPS stations (Figure 1) in the year 2020. This dataset represents an improvement

over the existing operational GPS PWV dataset provided by the Nevada Geodetic Laboratory (Yuan et

al., 2023). To enhance accuracy, meteorological data from the fifth generation of European ReAnalysis

(ERA5) with a higher spatio-temporal resolution is employed to correct the zenith hydrostatic delay of

the GPS signal. The mean absolute bias and standard deviation of the enhanced PWV dataset have been

reduced by 19.5% and 6.2%, respectively, in comparison to the operational dataset. Furthermore, the

incorporation of corrections derived from ERA5 has led to a notable decline by 92.4% in the prevalence

of implausible negative GPS PWV estimates.

Another GPS PWV product and a radiosonde PWV product were used to independently validate MWRI-

derived PWV. SuomiNet is a global network of GPS stations that provides high-quality PWV data with

an accuracy of better than 2 mm (Heise et al., 2009). Half-hourly SuomiNet PWV measurements from

stations worldwide, spanning the period from 2012 to 2020, were used. The Integrated Global

Radiosonde Archive Version 2 (IGRA2) represents the most comprehensive radiosonde dataset currently

available (Durre et al., 2018). The IGRA2 provides regular observations of atmospheric pressure, air

temperature, and humidity profiles at 00:00 and 12:00 UTC on a daily basis from over 2,800 stations

across the globe. Integration of the IGRA2 profiles of air temperature, pressure, and dew point from the

surface to a minimum of 500 hPa is employed to calculate PWV. Validation indicated the presence of a

5% dry bias associated with the estimation of PWV based on radiosonde data, in comparison to

observations obtained from well-calibrated microwave radiometers. IGRA2 PWV data in 2020 were used

here to validate MWRI PWV.

**2.3 Data collocation and ML algorithm development**

The development of the ML algorithm for PWV retrieval began with the collocation of GPS PWV values

with MWRI Tb measurements. To ensure reliability, the spatial distance between the GPS station and

the MWRI pixel was restricted to within 10 km, and the temporal difference was limited to under 15

minutes. This process yielded over one million data points. By leveraging this extensive dataset, which

encompasses a wide range of atmospheric and surface conditions, the ML algorithm was able to

effectively learn the complex relationship between MWRI Tb measurements and GPS PWV values,

facilitating accurate PWV retrieval from the MWRI data. Of the matched MWRI and GPS PWV data

points, 80% were used for training the ML algorithm, while the remaining 20% were reserved for

validation.



The efficacy of the ML algorithm is highly dependent on the engineering of features, the selection of estimators, and the tuning of hyperparameters. MWRI-based Tbs and their derivatives are used to depict pertinent information pertaining to atmospheric and surface conditions. The MAWVI is employed as a

primary feature as it is closely linked to PWV in physics. The ratio of horizontally polarized Tb at 23.8 GHz to that at 18.7 GHz, as well as the ratio of horizontally polarized Tb to vertically polarized Tb at 18.7 GHz, are also employed as features. These features have been demonstrated to be effective in representing vegetation transmissivity and open water fraction, respectively (Jones et al., 2010). The results of the SHAP analysis indicate that these features are of primary importance, suggesting that they

significantly contribute to the retrieval process (Xia et al., 2023a). The vertically polarized Tb at 36.5 GHz serves as a proxy for surface temperature (Ts), while cloud liquid water (CLW) is represented by the ratio of Tb difference at 89 GHz and 36.5 GHz. Auxiliary information, including station elevation, time, and location, is incorporated into the ML model as features (Xia et al., 2023a).

In order to facilitate the optimization and training of models in an efficient manner, the Fast and

Lightweight AutoML (FLAML) framework is utilized (Wang et al., 2021). FLAML expedites experimentation and enables efficient model optimization through the automated execution of tasks such as parameter optimization, model selection, dataset size determination, and runtime optimization. FLAML operates with two layers: the ML layer and the AutoML layer. The ML layer comprises a number of candidate models, including Light Gradient Boosting Machine (LightGBM) (Ke et al., 2017),

Extreme Gradient Boosting (XGBoost) (Chen and Guestrin, 2016), and Random Forest (Belgiu and Drăguţ, 2016). The AutoML layer is comprised of a number of components, including a model proposer, a hyperparameter and sample size proposer, a validation strategy proposer, and a controller. By leveraging FLAML, the PWV retrieval algorithm benefits from accelerated training, simplified model optimization, and enhanced efficiency in selecting the optimal model configuration. This comprehensive

approach guarantees the retrieval of robust and accurate PWV data.

Once the ML model has been trained, MWRI Level 1C (L1C) Tb data from FY-3 satellites (FY-3B: 2012-2017, FY-3D: 2018-2020), along with auxiliary information, are used to drive the model to produce pixel-level PWV estimates. These pixel estimates are then projected onto a gridded format with a spatial resolution of $0.25° \times 0.25°$ using a nearest neighbor interpolation method. To ensure data reliability, grids





with fewer than three PWV retrievals are excluded, guaranteeing that each grid cell has a sufficient number of data points for robust analysis. The resulting product is a global terrestrial gridded MWRI PWV dataset spanning from 2012 to 2020. This dataset provides a comprehensive coverage of PWV distributions across different terrestrial regions, facilitating in-depth analysis and insights into atmospheric water vapor dynamics on a global scale.

**3 Results**

**3.1 Overview evaluation of MWRI PWV against ground measurements**

Figure 2 presents the validation results for both the training and test datasets, using enGPS PWV as the benchmark. The MWRI PWV showed a strong correlation with enGPS PWV in the test set (Figure 2b), demonstrated by a coefficient of determination ($R^2$) of 0.96, a root mean square error (RMSE) of 2.27

mm, a mean bias error (MBE) of -0.02 mm, and a relative root mean square error (RRMSE) of 0.14. Additionally, the regression analysis between MWRI and enGPS PWV yielded a slope of 0.95 and an intercept of 0.71, indicating a slight dry bias in the MWRI measurements for higher PWV values.

Figure 3 shows the independent validation results against SuomiNet GPS PWV and IGRA2 PWV. When SuomiNet GPS PWV is used as a reference, the dataset achieves an $R^2$ of 0.85, an RMSE of 4.47 mm,

an MBE of 0.02 mm, and an RRMSE of 0.29. The linear regression analysis yields a slope of 0.95 and an intercept of 0.76, suggesting that the MWRI PWV tends to underestimate high PWV values. When IGRA2 PWV is used as the reference, the RMSE is 3.89 mm, the MBE is -0.32 mm, and the RRMSE is 0.76. The slope and intercept of the linear regression are 0.86 and 1.72, respectively, indicating a more pronounced dry bias in the MWRI PWV compared to IGRA2.

Figure 4 presents the validation results of the MWRI PWV across various surface types. The RMSE values for areas near water bodies (WAT and WET) are 4.43 mm and 3.69 mm, respectively. In forested regions (ENF, EBF, DNF, DBF, and MF), the RMSE ranges from 2.9 to 5.49 mm. This performance is notable, as previous passive microwave-based terrestrial PWV inversion algorithms struggled in these areas. Overall, the MWRI PWV product demonstrates satisfactory accuracy across all surface types,

proving effective even in complex conditions such as those near water and in densely vegetated regions.

In view of the discrepancies between the validation outcomes of the three ground-based PWV data sources (enGPS, SuomiNet GPS, and IGRA2), Figure 5 shows the probability density function (PDF) of PWV matches for MWRI versus SuomiNet (left), IGRA2 (middle), and enGPS (right) sites, respectively (upper panel), the PDF of PWV difference (middle panel), and the distribution of PWV differences (bottom panel). Compared to the three surface observation datasets, MWRI shows a higher proportion of PWV estimates within the 5-20 mm range and a lower proportion within the 0-5 mm and >20 mm ranges. This difference is the primary source of the discrepancy between MWRI and the ground-based observation datasets. For the PDF of the PWV differences between MWRI and SuomiNet as well as IGRA2, approximately 90% of the differences fall within the -3 to 3 mm range. Specifically, 45% of the differences between MWRI and SuomiNet are within the -1 to 1 mm interval, while only 35% of the differences between MWRI and IGRA2 fall within this interval. In contrast, the PDF of the PWV differences between MWRI and enGPS is more concentrated, with around 55% of the differences in the -1 to 1 mm range and only 2% exceeding 5 mm. To analyze the distribution of differences in each PWV interval, box plots of PWV differences were created for each 5 mm bin. The results reveal that MWRI PWV shows a wet bias in the 0-15 mm range compared to the three surface PWV products. Conversely, a notable dry bias is observed for PWV values exceeding 20 mm, especially at higher PWV levels. This trend reinforces the previous finding that MWRI tends to underestimate PWV when values are high. Compared to enGPS, MWRI shows a smaller tendency to underestimate, with an overall underestimation of less than 2 mm across all PWV intervals. However, MWRI tends to underestimate more than SuomiNet and IGRA2, with discrepancies exceeding 5 mm. The discrepancies observed in the validation against enGPS, SuomiNet GPS, and IGRA2 are likely due to inherent differences among these ground-based datasets. Figure 6 compares enGPS with SuomiNet GPS (left) and IGRA2 (right) at collocated sites. Although enGPS generally agrees well with both SuomiNet and IGRA2, notable differences persist. The RMSE is 3.8 mm between enGPS and SuomiNet and increases to 4.8 mm when comparing enGPS with IGRA2. Additionally, IGRA2 tends to underestimate PWV, especially at higher values, which explains why MWRI measurements align more closely with IGRA2 than with enGPS at elevated PWV levels.





To further explore the spatial characteristics of the MWRI PWV error, the PWV differences between MWRI PWV and PWV measured at each SuomiNet and IGRA2 station are calculated and shown in the form of spatial distribution in Figure 7. Overall, RMSE ranges from 2 to 6 mm. Most of the stations with high RMSE values (dark red) are located in the mid- to low-latitude Marine Annex region where water vapor is abundant. In contrast, stations with lower to moderate RMSE values are primarily found in polar regions and on continents. Relative errors are generally lower in the Southern Hemisphere, with a RRMSE around 0.2, compared to the Northern Hemisphere, where RRMSEs are between 0.3 and 0.4, particularly in North America and continental Europe. A few sites with unusual errors were examined in more detail (Figure 8). Although there is a notable discrepancy in PWV retrievals between MWRI and SuomiGPS (Figures 8b and 8d) as well as between MWRI and IGRA2 (Figure 8f), MWRI and enGPS still show a good agreement in PWV values. This suggests that the larger RMSE values when using SuomiNet/IGRA2 PWV as references, rather than enGPS PWV, are likely due to the inherent differences in PWV retrievals between enGPS and SuomiNet/IGRA2.

## 3.2 Potential geographical dependence of MWRI PWV retrievals

The influence of latitude, elevation, and surface skin temperature on the accuracy of MWRI PWV retrievals is further analyzed. Figure 9 presents a heat map depicting the relative differences in PWV between MWRI and enGPS, correlated with latitude (Figure 9a), surface skin temperature (Figure 9b), and elevation (Figure 9c). These relative differences are calculated within specific PWV percentiles to reduce the impact of PWV variations on the observed agreement. A clear dependence on latitude is observed, with an overall PWV overestimation observed in the 0~50th percentile latitude interval. This overestimation increases with decreasing latitude, reaching a relative deviation of 1.01 at some stations near the equator where. However, at higher latitudes, this tendency reverses, with an underestimation of PWV values becoming more prevalent. Overall, the magnitude of underestimation is less than that of overestimation, with MWRI PWV exhibiting a larger overall error in regions where PWV values are higher. Regarding the dependence on surface skin temperature, the distribution of biases shows that both negative (blue) and positive (red) biases are generally aligned along a diagonal line. The relative bias increases gradually as surface skin temperature and PWV decrease, ranging from -0.24 to 0.75. Biases are observed to be smaller in regions with higher surface skin temperatures and lower PWV values,



indicating that MWRI PWV tends to be underestimated in drier regions. Conversely, overestimation of MWRI PWV is more pronounced in areas with lower surface skin temperatures and higher PWV values. These results suggest that as surface skin temperature rises, MWRI PWV transitions from a dry bias to a wet bias, with this trend becoming more pronounced as PWV values increase. For elevation intervals

between the 0th and 50th percentiles, the relative deviation is predominantly positive, with only a few exceptions. As elevation decreases, the relative deviation increases gradually, reaching a maximum of 0.58. In contrast, at higher elevations, the relative deviation ranges from 0 to -0.17. The impact of elevation on relative deviation is relatively minor across different PWV intervals. These findings suggest that MWRI PWV tends to progressively overestimate PWV values as elevation decreases, and this

overestimation trend is largely unaffected by changes in PWV.

### 3.3 Assessment of long time series against AIRS and ERA5

Given that the MWRI BT data used to generate the terrestrial MWRI PWV product spans approximately ten years, it is crucial to evaluate the long-term stability of MWRI PWV. Since the enGPS PWV product covers only a single year, 2020, we utilized the SuomiNet GPS and IGRA2 PWV products as reference

data for the extended time series analysis.

Figure 10 displays the monthly mean PWV values from 2012 to 2020 for stations with SuomiNET observations in both the northern and southern hemispheres. Time series of ERA-5, AIRS, SuomiNET, and MWRI PWV products are compared. Overall, MWRI PWV exhibits a strong correlation with other products in terms of seasonal variations and aligns well with the interannual variations observed in

ground-based PWV measurements. The correlation coefficients range from 0.69 to 0.84, while the RMSEs vary between 2.74 and 6.44 mm. It is notable that the high RMSEs occur in the low-latitude sites (Figure 10(a), (c), (h)), while the RMSEs are relatively low for the mid- and high-latitude sites. ERA-5 PWV was found to have the highest overall accuracy and consistency compared with ground-based measurements, largely due to the assimilation of extensive ground-based site data. In contrast, while IRS

PWV showed strong agreement with ground-based measurements in terms of seasonal variability, it exhibited significant overall overestimation or underestimation of PWV at several sites (see Figures. 10(a), (b), and (h)). The MBE of AIRS at these sites is 3.38, 4.05, and -7.18 mm, respectively. It is noteworthy that in Figure 10(g), both ERA-5 and AIRS exhibit a considerable underestimation, with

MBEs of -3.13 and -3.27 mm, respectively. In comparison, the MBE of MWRI is only -1.00 mm,

demonstrating overall superior accuracy compared to these two products. In contrast, Figure 10(h)

illustrates that all three products significantly underestimate the summer PWV values. The MWRI

exhibits the least overall underestimation, with an MBE of -3.67 mm, while the MBEs for the ERA-5

and AIRS are -4.90 and -7.18 mm, respectively.

Figure 11 illustrates the global seasonal average PWV distributions from MWRI, ERA5, and AIRS for

the four seasons. Overall, the PWV distributions from these three products exhibit a clear latitudinal

dependence, with PWV decreasing as latitude increases. This pattern aligns with the Clausius-Clapeyron

equation, which indicates that warmer air at lower latitudes holds more water vapor compared to cooler

air at higher latitudes, except in arid regions like deserts. The three products show similar seasonal pattern

of PWV, with the regions of highest PWV values shifting north of the equator during the Northern

Hemisphere's summer months and south of the equator during the winter months. In the summer, the

areas with the highest PWV values are found in Southeast Asia, South Asia, northern South America,

and regions north of the equator, where PWV reaches 50-60 mm. Conversely, during the winter season,

the region of high PWV values shifts southward, accompanied by a decrease in precipitation in mid- and

high-latitude northern hemisphere regions, with PWVs decreasing to 5-15 mm. Note that discrepancies

remain in the PWV distributions of the MWRI, ERA5 and AIRS in certain regions, particularly during

the spring and summer months. During this period, the MWRI and AIRS exhibit a similar distribution.

In particular, during the spring and summer months, the MWRI PWV is observed to be approximately

3–5 mm higher than that of the ERA5 and AIRS in the Asian and European regions between 40°N and

60°N. Additionally, the centre of high PWV is noted to be approximately 5 mm lower than the ERA5

PWV and the AIRS PWV in central Africa. This finding is also consistent with the previous conclusion

that MWRI PWV exhibits a wet bias at low PWV values and a slight PWV underestimation at high PWV

values.

**3.4 MWRI PWV observations during an extreme weather event**

In addition to the well-documented spatial distribution and seasonal trends of water vapor, there is also

notable transport of water vapor to inland regions over short periods. This is due to the influence of



subtropical high pressure, tropical cyclones, and other abnormal weather events, which can cause significant and rapid changes in water vapor.

To assess whether MWRI PWV can promptly respond to rapid changes in water vapor over a short timescale, we analyze the PWV variation during the rainstorm that occurred in Beijing on August 12,

2020. On the afternoon of August 11, 2020, the Beijing Meteorological Bureau issued a yellow warning for heavy rainfall. The city experienced exceptionally heavy precipitation and the average rainfall ranged from 40 to 80 mm, with some areas receiving over 200 mm. The heavy rain persisted until August 13. Figure 12 illustrates the PWV distribution in the Beijing area from August 11 to August 13 (DOY 224–226) as observed by MWRI, AIRS, and ERA5. At 06:00 UTC on August 12 (DOY 225), a significant

increase in PWV values is evident in southern Beijing. The area of high PWV values rapidly expanded to encompass the southern, eastern, and northern parts of Beijing, with the maximum PWV reaching approximately 60 mm. At 24:00 UTC on August 12, the center of the high PWV area decreased in magnitude and shifted to the southeast. Figure 12(m)~(x) presents the PWV distributions of AIRS and MWRI in the ascending and descending orbits (corresponding to the early morning and afternoon of the

local time, respectively) around Beijing from 11 to 13 August. While the AIRS product is able to capture changes in PWV to a certain extent, its coarse resolution limits its portrayal of the water vapor distribution. Furthermore, the difficulty of IR in penetrating clouds resulted in the frequent lack of AIRS PWV retrievals . The overall PWV distribution of MWRI is similar to that of AIRS, but its higher resolution allows for a more subtle variation of PWV distribution. Furthermore, MWRI is not affected by clouds

and can provide all-weather PWV data. Figure 13 depicts the time series of regional mean PWV values around Beijing from 11 August to 13 August. Hourly ERA5 PWV values are used as the reference, with the PWV values during the overpass times of AIRS and MWRI (both ascending and descending orbits) represented by dots. Overall, MWRI PWV exhibits a very good consistency with the ERA5 PWV, with a difference of approximately 1–2 mm for the instantaneous observations. In contrast, the AIRS PWV

displays a more pronounced overall variation, with a discrepancy of up to 5 mm compared to the ERA5 PWV, and in some instances exceeding 10 mm. This notable divergence may be attributed to the lower resolution of the AIRS data and the potential for missing observations under cloudy conditions.





**4 Discussion**

Retrieving PWV over land using satellite passive microwave imagers, such as MWRI, is inherently
challenging due to the minimal temperature contrast between the Earth's surface and the atmosphere.
This renders the distinction between microwave signals emitted by the atmosphere and those emitted by
the surface a challenging endeavor. To address this issue, the study employs a data-driven retrieval
method that meticulously considers surface conditions, thereby enabling the effective detection of the
subtle water vapor signal. The validation of this approach demonstrates the efficacy of PWV retrieval
across diverse surface types, although the accuracy may vary depending on the specific surface
characteristics. The study underscores a greater consistency between MWRI and enhanced GPS PWV
products in comparison to SuomiNet GPS and IGRA2, emphasizing the necessity of acknowledging
potential discrepancies in PWV observations, even when utilizing analogous GPS-based retrieval
techniques. The MWRI method is primarily based on the use of a moderate water vapor absorption line
at 22.4 GHz, which presents certain challenges, particularly in dry conditions. In contrast, more recent
microwave imagers, such as the FY-3G, incorporate a stronger water vapor absorption line at 183 GHz,
which has the potential to enhance PWV retrieval over land. The incorporation of these advancements
has the potential to significantly enhance the accuracy and reliability of future PWV datasets.

The ten-year precipitable water vapor dataset over land developed in our study, derived from MWRI
measurements on FY3-B, FY3-C, and FY3-D, marks a significant advancement in remote sensing,
weather analysis, and climate science. Given the rarity of long-term datasets of this nature, this dataset
addresses a critical gap in the field. Researchers are expected to leverage it to explore the role of water
vapor in weather patterns, refine precipitation forecasting, and validate climate simulations. It will be
instrumental in detecting atmospheric rivers, understanding moisture distribution, and assessing its
effects on weather systems and climate. Moreover, the dataset is invaluable for hydrological models that
simulate the water cycle, aiding in water resource management, drought assessment, and flood risk
evaluation. Additionally, it provides a key reference for validating and improving other satellite-based
precipitable water vapor products, thereby enhancing the overall accuracy of satellite observations.

## 5 Summary and outlook


This study presents a global terrestrial all-sky PWV dataset from FY-3 MWRI, with $0.25° \times 0.25°$ spatial resolution and twice-daily temporal resolution (ascending and descending orbits) spanning 2012 to 2020. Validation of the MWRI PWV product reveals an RMSE of 2.27 mm when compared to enGPS PWV, while the RMSEs are 4.47 mm and 3.89 mm relative to SuomiNet GPS PWV and IGRA2 PWV,

respectively. Analysis of MWRI PWV errors shows distinct spatial distributions and dependencies on environmental conditions. Errors are significantly higher at low and mid-latitude stations, particularly in oceanic regions, whereas mid- and high-latitude stations, including polar regions, exhibit lower errors. MWRI tends to overestimate PWV at low latitudes, with a gradual shift toward underestimation at higher latitudes. This latitude-dependent bias highlights variability in PWV estimation across different

geographic zones. Overall, errors are lower in the Southern Hemisphere compared to the Northern Hemisphere. Additionally, as surface skin temperature increases, MWRI PWV errors shift from a dry bias (underestimation) to a wet bias (overestimation), with this effect becoming more pronounced at higher PWV values. The accuracy of MWRI PWV is also influenced by site elevation, showing a consistent trend of overestimation with decreasing elevation, irrespective of PWV values.

The MWRI PWV product effectively tracks seasonal variations in PWV, demonstrating overall superior performance compared to the AIRS PWV product and occasionally surpassing the ERA5 PWV product at individual ground-based stations. Its spatial patterns across different seasons closely align with those observed in ERA5 and AIRS, underscoring the reliability and versatility of MWRI PWV across various climatic conditions and geographical regions. An examination of an unusual weather event further

confirms that MWRI PWV can accurately capture rapid daily changes in PWV. In contrast, the coarse resolution of AIRS and potential cloud obscuration limit its ability to detect such PWV variations.

The MWRI PWV effectively tracks seasonal variations in PWV and demonstrates overall superior performance compared to the AIRS PWV product, occasionally surpassing the ERA5 PWV product at specific ground-based stations. Its spatial characteristics across different seasons closely match those

observed with ERA5 and AIRS, underscoring the reliability and versatility of MWRI PWV across various climatic conditions and geographic regions. An analysis of an unusual weather event further confirms that MWRI PWV accurately captured rapid daily changes in PWV, consistent with ERA5. In



contrast, the coarse resolution and susceptibility to cloud obscuration of AIRS limit its ability to capture such rapid PWV variations.

The MWRI PWV product overcomes the limitations of infrared observations. It provides accurate, all-weather PWV data over land, irrespective of surface type, offering a significant advancement in atmospheric research. Up until this point, the MWRI has been utilized to compile the dataset from only three satellites (FY3-B, FY3-C, and FY3-D). However, FY3-E, which was launched in 2021, does not carry an MWRI instrument and thus was not included in the dataset. Moreover, although two additional

FY3 satellites, FY3-F and FY3-G, were launched in 2023, their data are still undergoing testing. Consequently, measurements from these satellites have not been incorporated into our dataset as of now. In the future, we intend to expand the observational capabilities by incorporating data from satellites such as FY-3E, FY-3G, and FY-3F. This expansion will include coverage of morning, evening, and precipitation cycles, aiming to improve temporal resolution. These enhancements will provide more

detailed insights into the daily water vapor cycle and enable more responsive monitoring of short-term changes resulting from extreme weather events.



*Data availability.* The FY-3 MWRI level1 Tb data provided by NMSC are available at: https://satellite.nsmc.org.cn/PortalSite/Data/Satellite.aspx. The enhanced GPS PWV dataset provided by

Dr. Peng Yuan is available at: https://doi.org/10.5281/zenodo.6973528 (Yuan et al., 2022). The IGRA2

data were provided by National Centers for Environmental Information (NCEI) can be downloaded from

https://www.ncei.noaa.gov/access/metadata/landing-page/bin/iso?id=gov.noaa.ncdc:C00975.          The

SuomiNet GNSS dataset were provided by University Corporation for Atmospheric Research (UCAR)

and can be downloaded from https://www.cosmic.ucar.edu/what-we-do/data-processing-center/data. The

MWRI      PWV      dataset      described      in      this      paper      is      available      at:

https://doi.org/10.6084/m9.figshare.26712709 (Xia et al., 2024).

*Author contributions.* XRX: Conceptualization, Methodology, Formal analysis, Investigation, Software,

Writing original draft, and Visualization. RJ, MM: Investigation, Methodology. SW, PZ: Resources,

Validation. RJ, MM, SW, and PZ: Investigation and Reviewing. XAX: Investigation, Reviewing,

Supervision, Project administration, and Funding acquisition. All: Manuscript editing.

*Competing interests.* The contact author has declared that none of the authors has any competing interests.

Disclaimer. Publisher's note: Copernicus Publications remains neutral with regard to jurisdictional claims

in published maps and institutional affiliations.

*Acknowledgements.* We are grateful to NCEI and UCAR for providing IGRA2 and Suominet GPS PWV.

Sharing the enhanced GPS product by Dr. Peng Yuan is highly appreciated. We also thank NMSC for

providing the FY-3 MWRI level1 Tb data. The research is supported by National Science Foundation of

China (42075079).

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

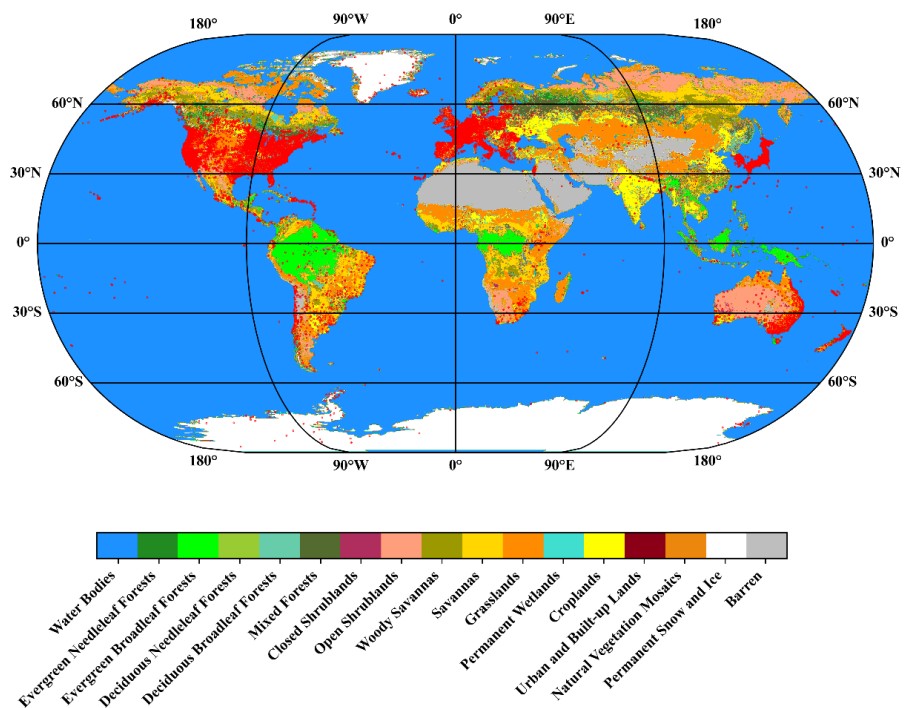

**Figure 1. Spatial distribution of 12552 enGPS PWV sites over the MODIS IGBP global land cover map, the PWV measurement from those sites are used for training of the FY-3 MWRI PWV retrieval ML model.**


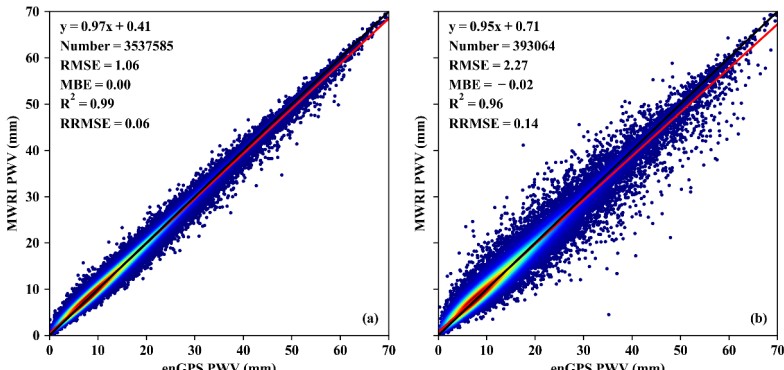

**Figure 2. Validation of FY-3 MWRI PWV in training set (a) and test set (b) using the enGPS PWV as a baseline.**


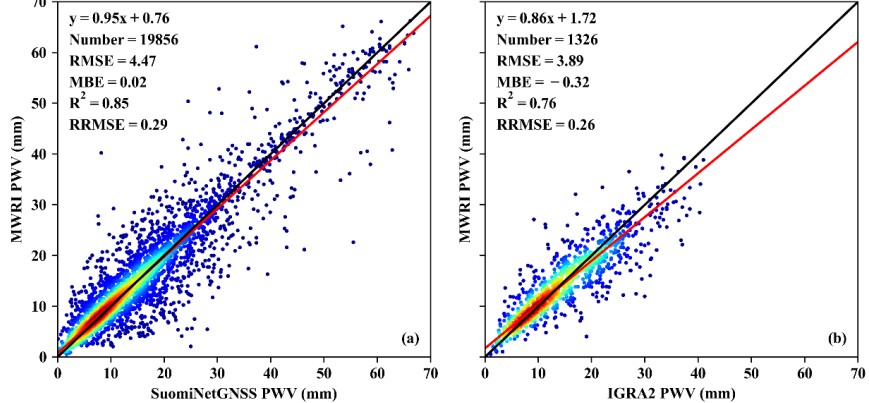

**Figure 3. Independ validation of FY-3 MWRI PWV using SuomiNet GNSS PWV (a) and IGRA2 PWV (b), respectively.**

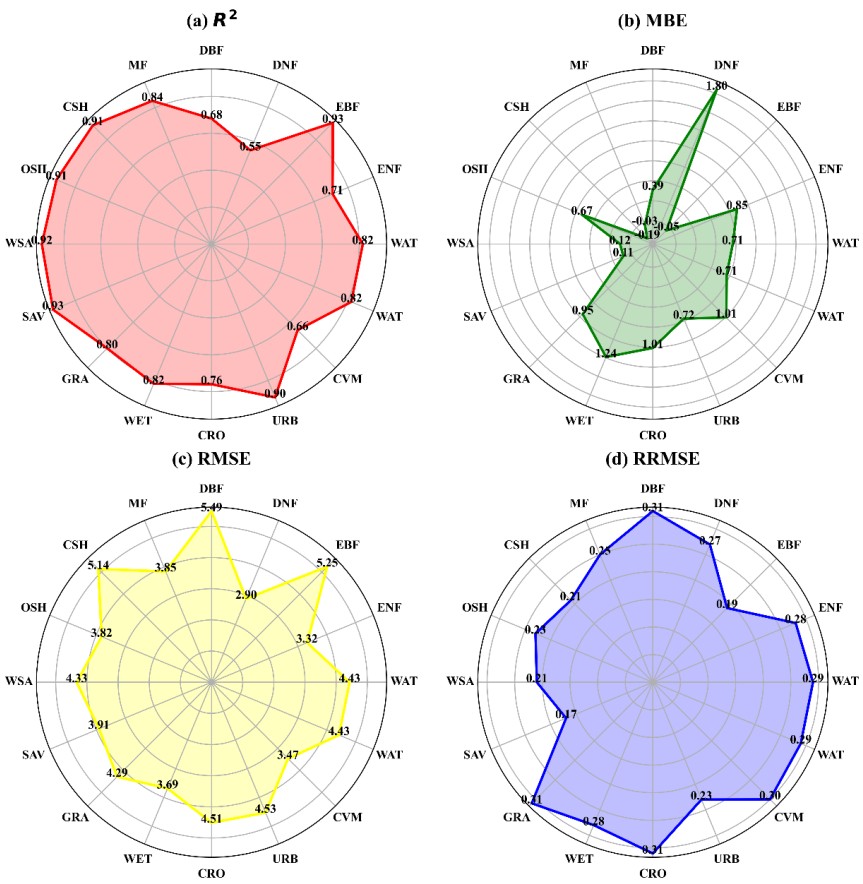

**Figure 4. Radar plot of (a) R², (b) MBE (mm), (c) RMSE (mm), (d) RRMSE of FY-3 MWRI PWV against SuomiNet GPS PWV and IGRA2 PWV in 14 IGBP types (Water Bodies (WAT), Evergreen Needleleaf Forests (ENF), Evergreen Broadleaf Forests (EBF), Deciduous Needleleaf Forests (DNF), Deciduous Broadleaf Forests (DBF), Mixed Forests (MF), Closed Shrubland (CSH), Open Shrublands (OSH), Woody Savannas (WSA), Savannas (SAV), Grasslands (GRA), Permanent Wetlands (WET), Croplands (CRO), Urban and Built-up Lands (URB), Natural Vegetation Mosaics (CVM)).**

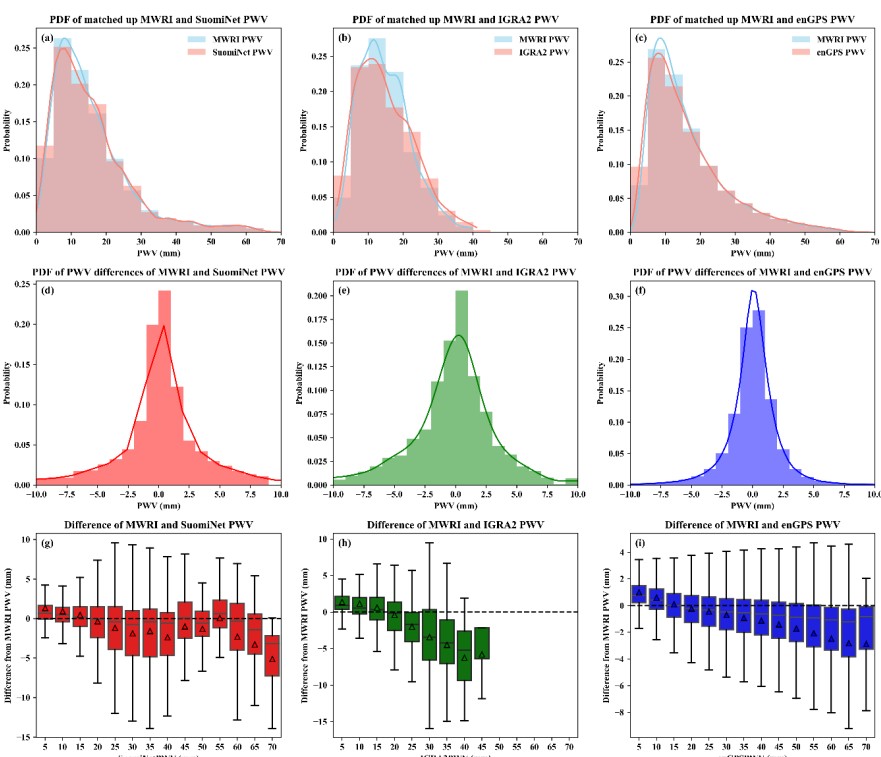

**Figure 5. The probability distribution function (PDF) of matched MWRI PWV and SuomiNet GPS PWV (a), IGRA2 PWV (b) and enGPS PWV (c) (top row). The PDF of PWV differences of MWRI PWV and SuomiNet PWV (d), IGRA2 PWV (e) and enGPS PWV (f) (middle row). Boxplot of PWV differences of MWRI PWV and SuomiNet PWV (g), IGRA2 PWV (h) and enGPS PWV (i) in each 5 mm PWV bins (bottom row).**

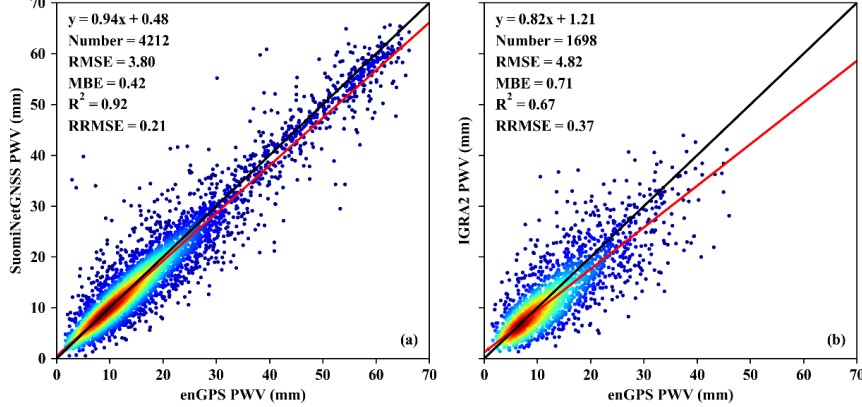

**Figure 6. Comparsion of PWV bewteen enGPS and SumiGPS (left) and between enGPS and IGRA2 (right)**


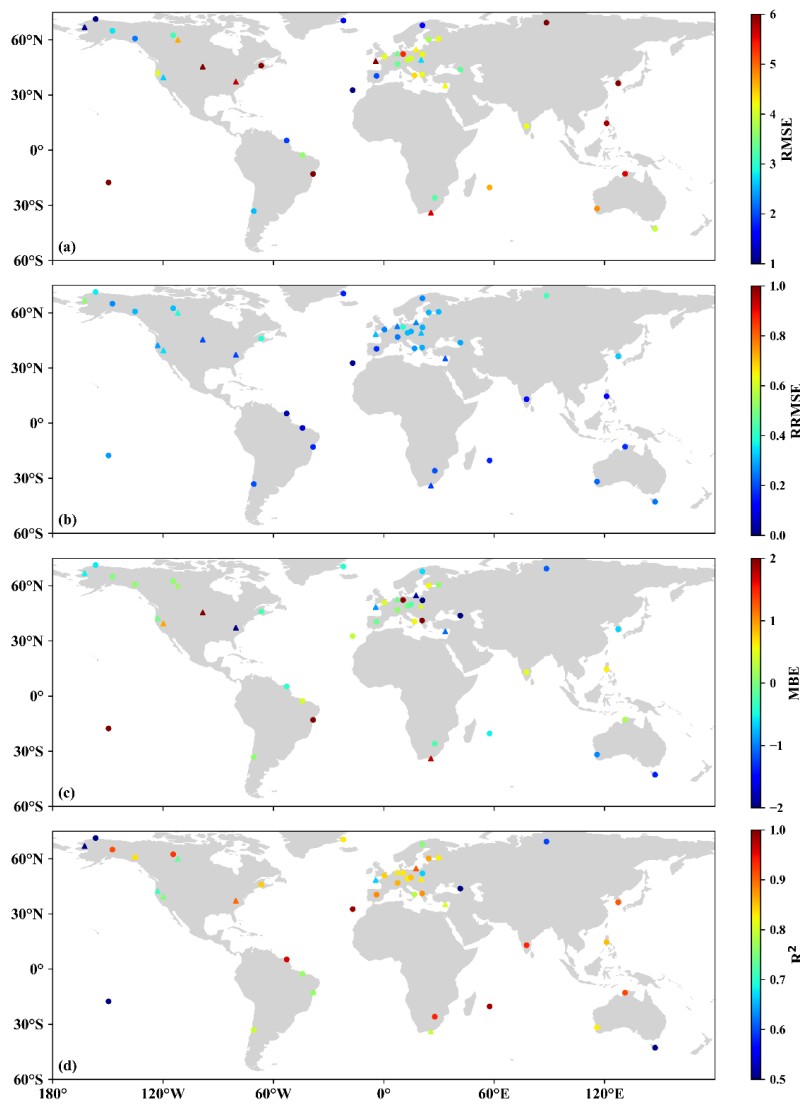

**Figure 7. Distribution of statistical metrics ( a: RMSE; b: RRMSE; c: MBE and d: R²) derived from comparison of MWRI PWV against SuomiNet (dots) and IGRA2 (triangles).**

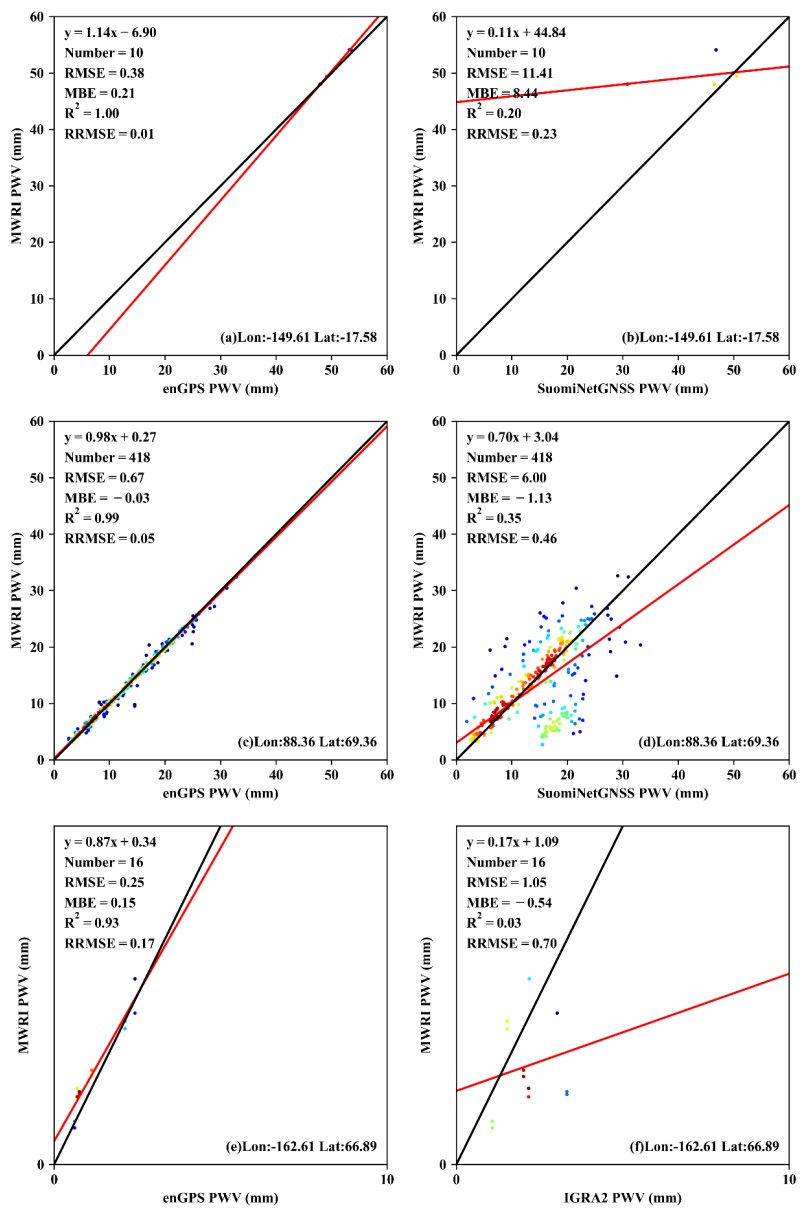

Figure 8. Comparison of PWV from MWRI against enGPS (right) for stations with abnormal differences between MWRI and SuomiNet PWV or IGRA2 PWV (RMSE > 7 mm or RRMSE > 0.4).



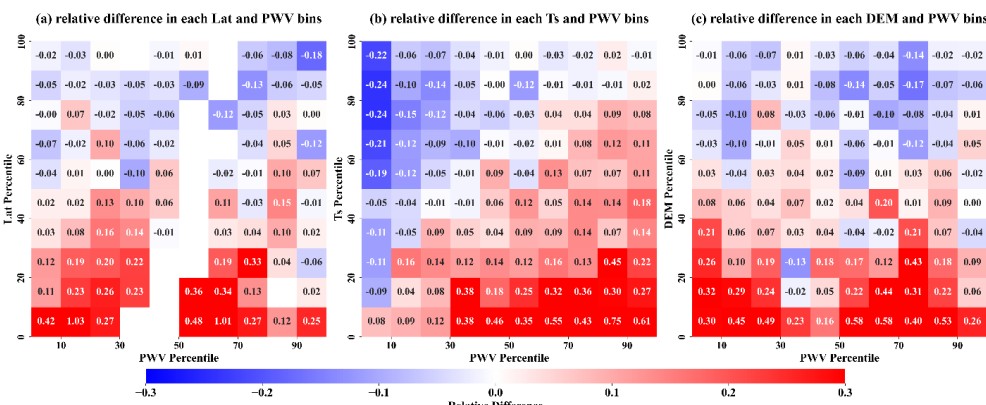

**Figure 9. Heatmap of relative differences in PWV between MWRI and enGPS across 10 percentile bins for Latitude (a), Surface Skin Temperature (b), and Elevation (c), analyzed within 10 percentile bins of PWV.**


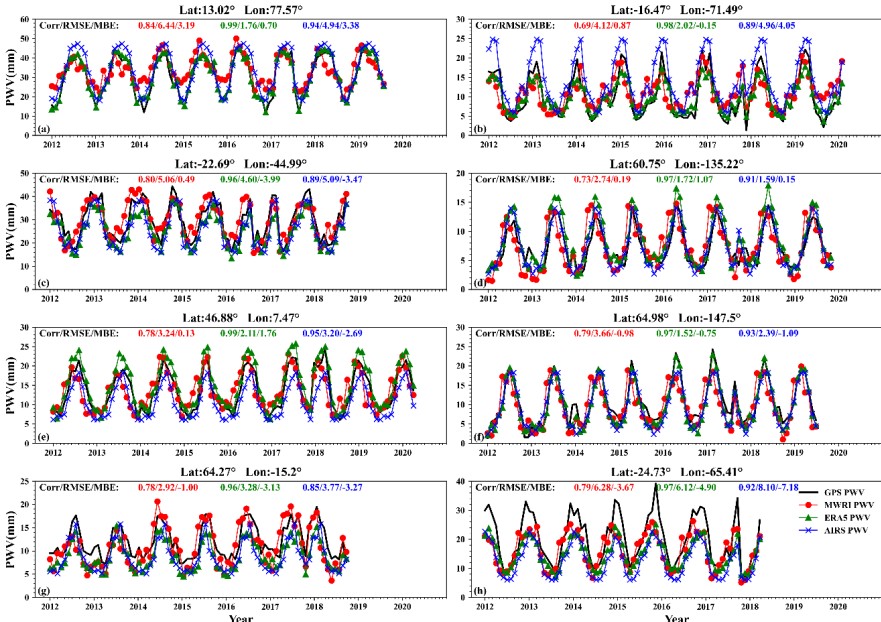

**Figure 10. The time series of monthly mean PWV values during 2021 to 2020 from MWRI, ERA5, AIRS and SuomiNet at stations with long-term SuomiNet PWV available.**


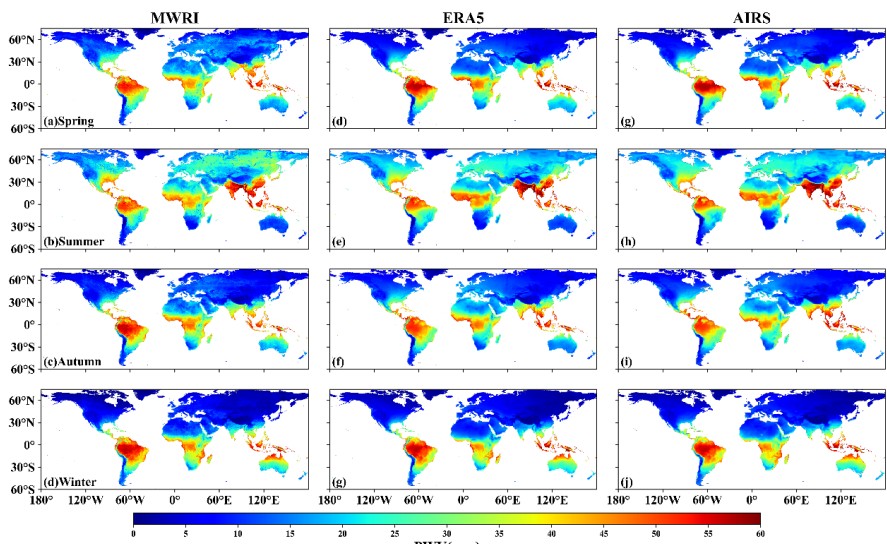

**Figure 11. Global seasonal average PWV over land from 2012 to 2020 (from top to bottom are spring, summer, autumn and winter), from left to right are 3 products: MWRI PWV, ERA-5 PWV and AIRS PWV, respectively.**



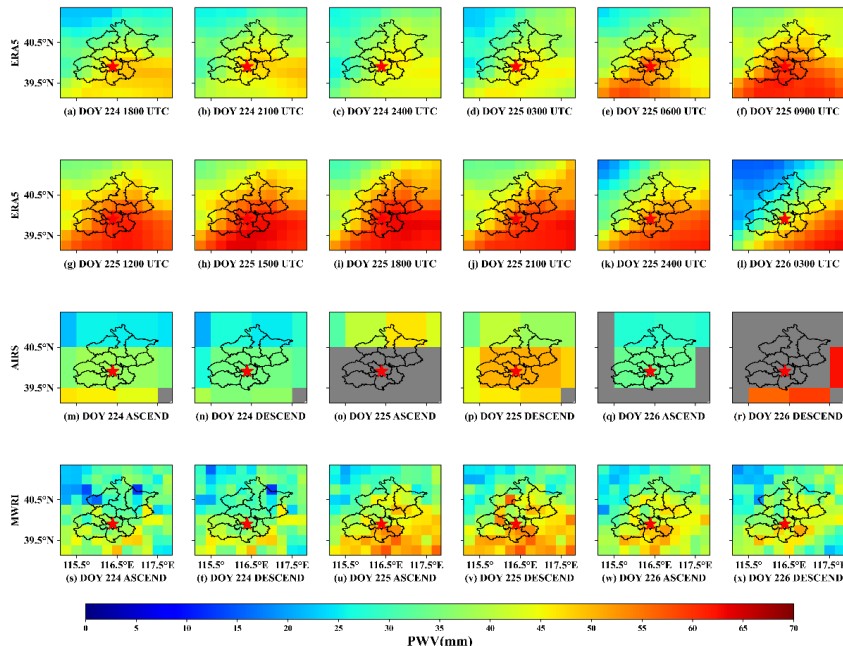

**Figure 12. The PWV distribution around Beijing from 2020.08.11 to 2020.08.13 (from day of year (DOY) 224 to DOY 226). The first two rows show the spatial distribution of PWV every 3 hours derived from ERA-5, and the third and fourth rows show the spatial distribution of PWV derived from AIRS and MWRI from the ascending and descending orbits, respectively.**



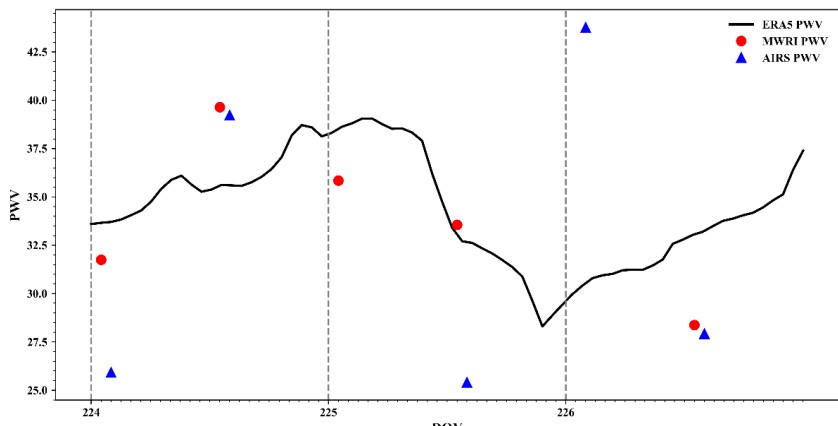


**Figure 13. The mean PWV in the Beijing region from 2020.08.11 to 2020.08.13 (from DOY 224 to DOY 226), where the black line is plotted from the hourly mean ERA5 PWV, while the red dots and blue triangles represent corresponding PWV from MWRI and AIRS, respectively.**