# Peer review of "A Global Terrestrial Precipitable Water Vapor Dataset from 2012 to 2020 Based on Microwave Radiation Imager Measurements from Three Fengyun Satellites"

_Earth System Science Data, 2024_

## Referee Comment (RC2)

**Review of " A Global Terrestrial Precipitable Water Vapor Dataset from 2012 to 2020 Based on Microwave Radiation Imager Measurements from Three Fengyun Satellites" by Xia et al.**

The authors developed a global terrestrial precipitable water volume (PWV) dataset from 2012 to 2020 by applying a machine learning model using Microwave Radiation Imager (MWRI) observations on board the Fengyun satellites series. The accuracy of dataset is evaluated by comparing with the products of SuomiNet GPS and Integrated Global Radiosonde Archive Version 2 (IGRA2) PWV. This work contributes to representing spatial and temporal PWV variations and providing valuable resource for atmospheric research. The manuscript may be considered for publication after being major revised in accordance with the following comments:

General:

1. The introduction could benefit from a more comprehensive discussion of the significance of PWV dataset in the context. This could include a brief overview of existing challenges and gaps in PWV dataset construction, and how this dataset addresses them. Additionally, the literature review should be expanded to include more recent studies on PWV retrieval employing machine learning techniques. This may help establish the novelty and contribution of approach proposed in this study.

2. In the method section, the authors choose Light Gradient Boosting Machine (LightGBM), Extreme Gradient Boosting (XGBoost) and Random Forest to train the model. The reasons for selecting these models should be supplemented.

3. In the conclusion, it is essential to articulate not only the strengths of the dataset but also to elucidate its constraints and limitations.

4. Please check the grammar in the manuscript to improve the text quality. For example, the subject of the sentence that "With the development of computer science, and in particular the proliferation of machine learning (ML), has led to the widespread adoption of ML by the remote sensing community" is missing.

Specific:

1. Line 216-217, The full names of "WAT, WET, ENF, EBF, DNF, DBF, and MF" should be provide when they first appear in the manuscript.

2. Figure 5: It is clear that the amount of data when MWRI PWV is compared to IGRA2 PWV is much smaller than when it is compared to SuomiNet GPS PWV and enGPS PWV. This should be supplemented in the manuscript as well as

giving possible reasons for this discrepancy.

3. Figure 8: The different colors of the solid dots in the figure should be clearly explained. Please include a color bar to indicate what each color represents for better clarity.

4. The manuscript states that "the MWRI PWV exhibits a wet bias at low PWV values and a slight PWV underestimation at high PWV values." Could you provide possible explanations for these observed biases? Discussing potential reasons, such as instrument limitations, atmospheric conditions, or retrieval algorithm issues, would help clarify this point.

---

## Author Comment (AC1)

Reply to the reviewer #1.

This paper employs a Lightweight Auto machine learning framework to produce a global terrestrial precipitable water vapor (PWV) dataset based on the MicroWave Radiation Imager (MWRI) aboard the FY-3 satellite series (FY-3B, FY-3C and FY-3D) spanning 2012 to 2020. The training dataset for the machine learning model is the enhanced GPS PWV dataset. SuomiNet GPS PWV, IGRA2 radiosonde, and the enhanced GPS PWV are used as reference datasets for validation. The authors examined the product quality from three perspectives: statistical fitting, spatial distribution, and temporal variation, while also assessing performance over different land surface types. It is recommended to be accepted after major revisions:

R: We sincerely appreciate your time and effort in reviewing our manuscript. Your valuable opinions and comments have been instrumental in improving the quality of our work. A point-by-point response is provided below, with the issues raised presented in black and our responses highlighted in blue.

1. The authors have not explained why the enhanced GPS PWV dataset was chosen as the training data for the machine learning model. This raises questions about the rationale of the research method.

R: Thank you for your thorough review. We choose the enhanced GPS PWV dataset as the training data for the following reasons: 1) The enhanced GPS PWV dataset provided an unprecedented number of GPS training samples (over 50 million) spanning diverse surface types from over 12,000 stations worldwide. 2) Compared to the operational GPS PWV product, the enhanced GPS PWV dataset demonstrates significant improvements in accuracy. Specifically, the mean absolute error (MAE) and standard deviation (SD) of the enhanced GPS PWV dataset, when compared against radiosonde-derived PWV, are reduced by an average of 19.5% and 6.2%, respectively. Furthermore, the number of unrealistic negative GPS PWV estimates is also significantly reduced by 92.4%, thanks to the accurate zenith hydrostatic delay (ZHD) derived from ERA5 (Yuan et al., 2023).

In summary, the enhanced GPS PWV dataset provides a more comprehensive, more representative and more accurate PWV product, which is essential for training machine learning models. Given the fact that the data volume and accuracy are critical requirements for machine learning, we decide to use this newly released GPS product as the learning label. As a matter of fact, this decision forms the foundation for developing an accurate and robust machine learning model, capable of reliably retrieving PWV under varying surface conditions.

2. Around line 245, the explanation for the bias between MWRI PWV and IGRA2 PWV is based on the argument that "the enhanced GPS PWV shares the same bias with IGRA2 PWV." This explanation lacks persuasive power and is not supported by relevant studies.

R: Thank you for your careful review. To clarify, our intention was to highlight that IGRA2 exhibited an underestimation of PWV at high PWV values when compared to MWRI PWV, and a similar trend was observed when comparing IGRA2 and enGPS PWV. The underlying reason for this phenomenon remains unclear and warrants further investigation. We apologize for this misunderstanding caused by our previous wording and have revised the manuscript accordingly to address this issue.

3. It is recommended to include an analysis of the machine learning model's uncertainty or error, particularly focusing on how the model performs under different weather conditions.

R: Thank you for your invaluable advice. We expanded our analysis by incorporating hourly ERA-5 total cloud cover and precipitation amount as the indicators of weather conditions. The matched ground-based PWV measurements (enGPS, SuomiNet, and IGRA-2) and MWRI PWV products are classified into six categories: those with precipitation and those without. The group without precipitation was further classified into four sub-classes based on CF (C1: CF ($<$ 0.1); C2: CF (0.1－0.3); C3: CF

(0.3 – 0.7); C4: CF (>0.7)) and the group with precipitation were classified into two sub-classes base on the amount of precipitation (C5: precipitation (0 – 5 mm) and C6: precipitation (>5 mm)). The RMSEs of MWRI PWV in the absence of precipitation range from 1.97 to 2.35 mm for different CF scenarios. While RMSE increases with higher CF, the overall uncertainty is still controlled within 2.35 mm. In cases of rainfall, the RMSE is 2.93 mm and 3.29 mm for the C5 and C6 scene, respectively. This result indicates that the MWRI PWV has a reliable performance under different weather conditions, although clouds and precipitation indeed reduce the accuracy of the MWRI PWV, but their overall effects are still tolerable.

[Figure]

**Figure 6. Evaluation of MWRI PWV under different weather conditions against ground-based PWV ((a) C1: CF < 0.1, (b) C2: 0.1 < CF < 0.3, (c) C3: 0.3 < CF < 0.7, (d) C4: CF > 0.7, (e) C5: precipitation < 5mm and (f) C6: precipitation > 5mm).**

4. The dataset performs poorly under extreme weather conditions. It is recommended to consider increasing the variety of training data for machine learning in such regions. By categorizing rainfall events, the authors could select the dataset that performs best under specific rainfall conditions as the training data for the machine learning model.

R: Thank you for your advice. As you suggested before, we evaluated the performance of our ML model under different weather conditions, the performance under extreme weather conditions, for example, heavy rainfall, deteriorates when compared to that under clear skies. This is understandable because it is very hard to fully account for the effect of rain droplets on microwave radiation (via scattering and absorption) under this situation. In other words, MWRI brightness temperature is not only influenced by PWV but also by highly variable rain droplets under this condition. Our goal is to develop a ML model capable of retrieving PWV from MWRI under all conditions, using only MWRI brightness temperature as input. Developing an independent ML model specifically for extreme weather conditions is a valuable suggestion and is worth considering in the future when we have more training data points.

5. The MWRI has a limited number of channels and lacks high-frequency channels, which makes it less sensitive to precipitation compared to sensors with high-frequency channels. Could this limitation be mitigated

by incorporating data from other FY-3 sensors?

R: Thank you for your advice. We are aware that MWRI only includes the 10.65~89 GHz channels. Channels in the 118 GHz and 183 GHz are more sensitive to precipitation and PWV. Micro-Wave HumiditySonder-2 (MWHS-2) onboard FY3C, FY3D, FY-3E and FY-3F satellites provide measurements at 118 GHz and 183 GHz. MWRI is a conical scanning imager, while MWHS-2 is an across-track scanning radiometer, both scanning techniques offer unique advantages. Combing these two techniques could potentially benefit from higher spatial resolution, improved retrieval accuracy and better coverage. However, effective data fusion and model development would be necessary to combine these two types of measurements in a meaningful way. It is also important to note that the goal of this study is to establish the longest PWV dataset using only MWRI and we also consider that MWHS-1 onboard FY3B does not provide 183 GHz data, so combing MWHS and MWRI is not feasible within this study. Nevertheless, this suggestion is valuable and we will consider incorporating additional channels in future to enhance the performance of the algorithm.

6. In Figure 7, the number of validation stations seems not enough, and the spatial distribution is uneven, with most stations concentrated in Europe. Is the validation in other regions reliable enough?

R: Thank you for your careful review. Indeed, most SuomiNet and IGRA-

2 stations are located in Europe and North America, and due to strict data collocation criteria (distance difference no more than 10 km, time difference no more than 15 min) between MWRI and ground-based data, many SuomiNet and IGRA-2 stations were excluded. To explore the uncertainty of MWRI PWV in other regions, we also validate MWRI PWV retrievals for six continents: Asia, Africa, North America, South America, Europe and Oceania. Figure 11 shows the comparison of MWRI PWV against PWV measurements from SuomiNet and IGRA-2 sites. MWRI PWV retrievals are reliable across all continents, although performance indeed varies between regions. The best results were observed in Africa and South America, despite the limited number of training data pints from these areas, as a large proportion of the training points comes from stations in Europe and North America. This variation in performance is likely due to differences in weather and surface conditions across continents.

[Figure]

**Figure 11. Taylor diagram of MWRI PWV against PWV driven by SuomiNet and IGRA-2 sites over 6 continents (Asia, Africa, North America, South America, Europe and Oceania).**

7. It is recommended to include a quality comparison between the FY-3 MWRI Level 1C Tb dataset and other Tb datasets to highlight the innovation of the study.

R: Thank you for your advice. Regarding the comparison of the FY-3 MWRI Level 1C Tb with other similar instruments in our previous work, we have conducted a comprehensive evaluation of the FY-3 MWRI channels over land and ocean, over ascending and descending orbits, using

the GPM GMI as a reference, and the results show that the bias of the MWRI in ascending and descending orbits after recalibration is well-controlled, with the overall MBE being less than 0.5 K and the RMSE being less than 1.5 K, respectively (Xia et al., 2023). We also highlighted this in Line 122 with: "Consequently, the precision of MWRI Tb datasets, particularly in the water vapor absorption channel, has been markedly enhanced. Cross-comparisons with datasets from other satellites, such as AMSR2 and GMI, have validated the effectiveness of the recalibrated MWRI Tb datasets (He et al., 2023; Xia et al., 2023b)" and Line 141 with: "Following the extensive reprocessing of FY-3 historical data, a new version of the long-term recalibrated FY-3 MWRI L1C Tb dataset has been released by NSMC (Wu et al., 2023). MWRI Tbs from 3 FY-3 satellites (FY-3B, FY-3C and FY-3D) were evaluated by using GMI as a reference, demonstrating that the newly recalibrated dataset exhibited a notable enhancement in accuracy, with the RMSE for each channel remaining below 2 K (Xia et al., 2023b)".

Xia, X., He, W., Wu, S., Fu, D., Shao, W., Zhang, P., and Xia, Xiangao: A Thorough Evaluation of the Passive Microwave Radiometer Measurements onboard Three Fengyun-3 Satellites, J. Meteorol. Res. 37, 573–588, https://doi.org/10.1007/s13351-023-2198-3, 2023.

8. Many of the references cited are outdated. It is recommended to

incorporate more recent studies in the literature review.

R: Thank you for your advice. We have added references to the more recent research advances in this direction that are currently available, as follows:

Zhao, Q., Ma, Z., Yin, J., Yao, Y., Yao, W., Du, Z., Wang, W.: General method of precipitable water vapor retrieval from remote sensing satellite near-infrared data. Remote Sensing of Environment 308, 114180. https://doi.org/10.1016/j.rse.2024.114180, 2024.

Zhou, S., Cheng, J.: A physics-based atmospheric precipitable water vapor retrieval algorithm by synchronizing MODIS near-infrared and thermal infrared measurements. Remote Sens. Environ. 317, 114523. https://doi.org/10.1016/j.rse.2024.114523, 2025.

Ma, X., Yao, Y., Zhang, B., He, C.: Retrieval of high spatial resolution precipitable water vapor maps using heterogeneous earth observation data. Remote Sensing of Environment 278, 113100. https://doi.org/10.1016/j.rse.2022.113100, 2022.

Jiang, N., Xu, Y., Xu, T., Li, S., Gao, Z.: Land Water Vapor Retrieval for AMSR2 Using a Deep Learning Method. IEEE Trans. Geosci. Remote Sensing 60, 1–11. https://doi.org/10.1109/TGRS.2022.3162222, 2022.

Jiang, N., Wu, Y., Li, S., Xu, Y., Wang, Y., Xu, T.: First PWV Retrieval Using MERSI‐LL Onboard FY‐3E and Cross Validation With Co‐Platform Occultation and Ground GNSS. Geophysical Research Letters 51, e2024GL108681. https://doi.org/10.1029/2024GL108681, 2024.

He, W., Chen, H., Xia, X., Wu, S., Zhang, P.: Evaluation of the Long-term Performance of Microwave Radiation Imager Onboard Chinese Fengyun Satellites. Adv. Atmos. Sci. 40, 1257–1268. https://doi.org/10.1007/s00376-023-2199-2, 2023.

Li, R., Hu, J., Wu, S., Zhang, P., Letu, H., Wang, Y., Wang, X., Fu, Y., Zhou, R., Sun, L.: Spatiotemporal Variations of Microwave Land Surface Emissivity (MLSE) over China Derived from Four-Year Recalibrated Fengyun 3B MWRI Data. Adv. Atmospheric Sci. 39, 1536–1560. https://doi.org/10.1007/s00376-022-1314-0, 2022.

Wang, Y., Jiang, N., Wu, Y., Xu, Y., Kaufmann, H., Xu, T.: An Improved Model for the Retrieval of Precipitable Water Vapor in All-Weather Conditions (RCMNT) Based on NIR and TIR Recordings of MODIS. IEEE Trans. Geosci. Remote Sens. 62, 1–12. https://doi.org/10.1109/TGRS.2024.3381750, 2024.

Su, H., Yang, T., Wang, K., Sun, B., Yang, X.: Evaluation of Precipitable Water Vapor Retrieval from Homogeneously Reprocessed Long-Term GNSS Tropospheric Zenith Wet Delay, and Multi-Technique. Remote Sensing 13, 4490. https://doi.org/10.3390/rs13214490, 2021.

He, J., Liu, Z.: Water Vapor Retrieval from MODIS NIR Channels Using Ground-Based GPS Data. IEEE Trans. Geosci. Remote Sensing 58 (5), 3726–37. https://doi.org/10.1109/TGRS.2019.2962057, 2020.

He, J., Liu, Z.: Refining MODIS NIR Atmospheric Water Vapor Retrieval

Algorithm Using GPS-Derived Water Vapor Data. IEEE Trans. Geosci.

Remote            Sensing            59,            3682–3694.

https://doi.org/10.1109/TGRS.2020.3016655, 2021.

---

## Author Comment (AC2)

Reply to the reviewer #2

The authors developed a global terrestrial precipitable water volume (PWV) dataset from 2012 to 2020 by applying a machine learning model using Microwave Radiation Imager (MWRI) observations on board the Fengyun satellites series. The accuracy of dataset is evaluated by comparing with the products of SuomiNet GPS and Integrated Global Radiosonde Archive Version 2 (IGRA2) PWV. This work contributes to representing spatial and temporal PWV variations and providing valuable resource for atmospheric research. The manuscript may be considered for publication after being major revised in accordance with the following comments:

R: Thank you for your patient review. Your insights and suggestions are extremely helpful in refining our manuscript. A detailed response to each of your points is outlined below, with the issues raised presented in black and our responses highlighted in blue.

General:

1. The introduction could benefit from a more comprehensive discussion of the significance of PWV dataset in the context. This could include a brief overview of existing challenges and gaps in PWV dataset construction, and how this dataset addresses them. Additionally, the literature review should be expanded to include more recent studies on PWV retrieval employing machine learning techniques. This may help establish the novelty and

contribution of approach proposed in this study.

R: Thank you for your advice. We revised the manuscript according to your invaluable suggestions. 1) To demonstrate the importance of the PWV dataset, in line 60, we added: "With all-weather global PWV records, researchers are expected to use them to study the role of PWV in weather patterns, refine precipitation forecasts, and validate climate simulations". In line 414 we added: "It will be instrumental in detecting atmospheric rivers, understanding moisture distribution, and assessing its effects on weather systems and climate. Moreover, the dataset is invaluable for hydrological models that simulate the water cycle, aiding in water resource management, drought assessment, and flood risk evaluation. Additionally, it provides a key reference for validating and improving other satellite-based precipitable water vapor products, thereby enhancing the overall accuracy of satellite observations". 2) For the recent advances in PWV retrieval based on machine learning techniques. in line 84 we added: "Jiang et al. (2022) developed a back-propagation neural network (BPNN) to retrieve PWV over land with the RMSE of 3.87 mm". We also cite more related articles up to date as follow:

Zhou, S., Cheng, J.: A physics-based atmospheric precipitable water vapor retrieval algorithm by synchronizing MODIS near-infrared and thermal infrared measurements. Remote Sens. Environ. 317, 114523. https://doi.org/10.1016/j.rse.2024.114523, 2025.

Ma, X., Yao, Y., Zhang, B., He, C.: Retrieval of high spatial resolution precipitable water vapor maps using heterogeneous earth observation data. Remote Sensing of Environment 278, 113100. https://doi.org/10.1016/j.rse.2022.113100, 2022.

Jiang, N., Xu, Y., Xu, T., Li, S., Gao, Z.: Land Water Vapor Retrieval for AMSR2 Using a Deep Learning Method. IEEE Trans. Geosci. Remote Sensing 60, 1–11. https://doi.org/10.1109/TGRS.2022.3162222, 2022.

Jiang, N., Wu, Y., Li, S., Xu, Y., Wang, Y., Xu, T.: First PWV Retrieval Using MERSI-LL Onboard FY-3E and Cross Validation With Co-Platform Occultation and Ground GNSS. Geophysical Research Letters 51, e2024GL108681. https://doi.org/10.1029/2024GL108681, 2024.

He, W., Chen, H., Xia, X., Wu, S., Zhang, P.: Evaluation of the Long-term Performance of Microwave Radiation Imager Onboard Chinese Fengyun Satellites. Adv. Atmos. Sci. 40, 1257–1268. https://doi.org/10.1007/s00376-023-2199-2, 2023.

Li, R., Hu, J., Wu, S., Zhang, P., Letu, H., Wang, Y., Wang, X., Fu, Y., Zhou, R., Sun, L.: Spatiotemporal Variations of Microwave Land Surface Emissivity (MLSE) over China Derived from Four-Year Recalibrated Fengyun 3B MWRI Data. Adv. Atmospheric Sci. 39, 1536–1560. https://doi.org/10.1007/s00376-022-1314-0, 2022.

Wang, Y., Jiang, N., Wu, Y., Xu, Y., Kaufmann, H., Xu, T.: An Improved Model for the Retrieval of Precipitable Water Vapor in All-Weather

Conditions (RCMNT) Based on NIR and TIR Recordings of MODIS. IEEE Trans. Geosci. Remote Sens. 62, 1–12. https://doi.org/10.1109/TGRS.2024.3381750, 2024.

Su, H., Yang, T., Wang, K., Sun, B., Yang, X.: Evaluation of Precipitable Water Vapor Retrieval from Homogeneously Reprocessed Long-Term GNSS Tropospheric Zenith Wet Delay, and Multi-Technique. Remote Sensing 13, 4490. https://doi.org/10.3390/rs13214490, 2021.

He, J., Liu, Z.: Water Vapor Retrieval from MODIS NIR Channels Using Ground-Based GPS Data. IEEE Trans. Geosci. Remote Sensing 58 (5), 3726–37. https://doi.org/10.1109/TGRS.2019.2962057, 2020.

He, J., Liu, Z.: Refining MODIS NIR Atmospheric Water Vapor Retrieval Algorithm Using GPS-Derived Water Vapor Data. IEEE Trans. Geosci. Remote Sensing 59, 3682–3694. https://doi.org/10.1109/TGRS.2020.3016655, 2021.

2. In the method section, the authors choose Light Gradient Boosting Machine (LightGBM), Extreme Gradient Boosting (XGBoost) and Random Forest to train the model. The reasons for selecting these models should be supplemented.

R: Thank you for your advice. We added the reason why we chose the three ML models in Line 198 with: "Among the shallow network structures, tree-based models have been consistently shown superior performance. Briefly,

RF is an ensemble learning method that combines the outputs of multiple basic decision trees to make final predictions. Each decision tree is built by recursively partitioning the data based on the value ranges of various features. RF models have advantages in dealing with high-dimensional data, outliers and missing data (Belgiu and Drăguţ, 2016; Lundberg et al., 2020). XGBoost is an ensemble learning framework designed to construct an ensemble of weak decision trees that are combined using the gradient boosting technique. Each successive tree corrects the discrepancies between the prediction of the previous tree and the target value. By incorporating regularization techniques to prevent overfitting, XGBoost has gained popularity for to its high performance and reliability (Chen and Guestrin, 2016). LGBM is another gradient boosting framework that aims to provide faster training speed and lower memory consumption compared to other frameworks. It incorporates a technique called gradient-based one-sided sampling to select the most informative samples during the tree-building process. In addition, histogram-based gradient estimation, which takes advantage of binning for efficient computation, is used (Ke et al., 2017)".

3. In the conclusion, it is essential to articulate not only the strengths of the dataset but also to elucidate its constraints and limitations.

R: Thank you for your comments. To summarize the shortcomings of our

product and what can be improved in the future, we have added line 454 with: "The MWRI PWV retrievals are still improved under extreme precipitation events, which may be resolved to some extent by combing MWHS measurements with much more channels".

4. Please check the grammar in the manuscript to improve the text quality. For example, the subject of the sentence that "With the development of computer science, and in particular the proliferation of machine learning (ML), has led to the widespread adoption of ML by the remote sensing community" is missing.

R: Thank you for your careful review. We have replaced line 83~84 with: "With advancements in computer science, particularly the proliferation of machine learning (ML), ML has been widely adopted by the remote sensing community". At the same time, grammatical and expression errors elsewhere in the manuscript are being progressively corrected.

Specific:

1. Line 216-217, The full names of "WAT, WET, ENF, EBF, DNF, DBF, and MF" should be provide when they first appear in the manuscript.

R: Thank you for your careful review. We added the full names in Line 243~246 with: "(Water Bodie (WAT) and Permanent Wetlands (WET) are 4.43 mm and 3.69 mm, respectively. In forested regions (Evergreen

Needleleaf Forest (ENF), Evergreen Broadleaf Forest (EBF), Deciduous Needleleaf Forest (DNF), Deciduous Broadleaf Forest (DBF) and Mixed Forests (MF)), the RMSE ranges from 2.90 to 5.49 mm". We also added the full names of MODIS IGBP in Figure 1. (line 582) with: "(Water Bodie (WAT), Evergreen Needleleaf Forest (ENF), Evergreen Broadleaf Forest (EBF), Deciduous Needleleaf Forest (DNF), Deciduous Broadleaf Forest (DBF), Mixed Forests (MF), Closed Shrubland (CLS), Open Shrubland (OSH), Woody Savanna (WSA), Savanna (SAV), Grassland (GRA), Permanent Wetlands (WET), Croplands (CRO), Urban and Built-up Lands (URB), Natural Vegetation Mosaic (NVM), Permanent Snow and Ice (SNW), Barren (BDR)"

2. Figure 5: It is clear that the amount of data when MWRI PWV is compared to IGRA2 PWV is much smaller than when it is compared to SuomiNet GPS PWV and enGPS PWV. This should be supplemented in the manuscript as well as giving possible reasons for this discrepancy.

R: Thanks. For the amount of data MWRI PWV compared to IGRA2 PWV is much smaller than when it is compared to SuomiNet GPS PWV and enGPS PWV, the main reason is that IGRA-2 provided only twice a day (00:00 and 12:00 UTC), while GNSS can provide PWV with higher temporal resolution. Given the fact we match the satellite and ground-based PWV data points if the temporal difference between them should be

less than 15 minutes, we can obtain more MWRI-GPS pairs over the same time period. We added further explain in line 238 with: "Limited by the frequency of IGRA-2 measurements of PWV, we obtained a small sample size of MWRI and IGRA-2 matches."

3. Figure 8: The different colors of the solid dots in the figure should be clearly explained. Please include a color bar to indicate what each color represents for better clarity.

R: Thank you for your comments. Given that the use of a density scatterplot is not appropriate for a single site with a small number of samples, we have replaced Figure 10 with the following scatterplot to allow the reader to understand it more intuitively.

[Figure]

**Figure 10. Comparison of PWV from MWRI against enGPS (right) for stations with abnormal differences between MWRI and SuomiNet PWV or IGRA2 PWV (RMSE > 7 mm or RRMSE > 0.4).**

4. The manuscript states that "the MWRI PWV exhibits a wet bias at low PWV values and a slight PWV underestimation at high PWV values." Could you provide possible explanations for these observed biases? Discussing potential reasons, such as instrument limitations, atmospheric conditions, or retrieval algorithm issues, would help clarify this point.

R: Thank you for your valuable advice. The reason why the MWRI PWV shows a wet bias at low PWV values and a slight PWV underestimation at high PWV values can be caused by the following factors: 1) We currently lack a feature fully describing extreme PWV conditions (generally associated with rainfall) in our machine learning model. 2) The training samples with extremely lower or higher PWV values are still limited. These reasons would lead to an overestimation or underestimation of extreme dry or wet PWV events in our trained models. In the future, we will try to include more representative PWV samples (e.g. droughts, exceptionally heavy rainfall) to improve the accuracy of the model when more training data points are available.